# On Numerosity of Deep Neural Networks

**Xi Zhang**
Shanghai Jiao Tong University
zhangxi_19930818@sjtu.edu.cn

**Xiaolin Wu**∗
McMaster University
xwu@ece.mcmaster.ca

## Abstract

Recently, a provocative claim was published that number sense spontaneously emerges in a deep neural network trained merely for visual object recognition. This has, if true, far reaching significance to the fields of machine learning and cognitive science alike. In this paper, we prove the above claim to be unfortunately incorrect. The statistical analysis to support the claim is flawed in that the sample set used to identify number-aware neurons is too small, compared to the huge number of neurons in the object recognition network. By this flawed analysis one could mistakenly identify number-sensing neurons in any randomly initialized deep neural networks that are not trained at all. With the above critique we ask the question what if a deep convolutional neural network is carefully trained for numerosity? Our findings are mixed. Even after being trained with number-depicting images, the deep learning approach still has difficulties to acquire the abstract concept of numbers, a cognitive task that preschoolers perform with ease. But on the other hand, we do find some encouraging evidences suggesting that deep neural networks are more robust to distribution shift for small numbers than for large numbers.

## 1 Introduction

In the past decade deep convolutional neural networks (DCNN) have rapidly developed into a powerful problem-solving paradigm that has found a wide gamut of applications, spanning almost all academic disciplines. DCNNs trained by large data, or deep learning (DL) as known colloquially, are noted for their apparent intelligent behaviours, being able to solve difficult problems of pattern analysis, recognition and classification [1, 2, 3, 4, 5, 6, 7]. Although artificial neural networks were originally inspired by the knowledge of animal cortex [8], our understanding of the inner working mechanism of DL is outstepped or arguably even mystified by their functional prowess in many applications.

It is therefore interesting to differentiate and contrast DL machines and humans in their capabilities to perform primitive cognitive tasks. In this context, numerosity is an ideal Turing-type test on the cognitive power or deficit of DL. Numerosity or the awareness of numbers is a neurocognitive function possessed by human infants prior to speech and any symbolic learning and even by animals [9, 10, 11, 12, 13, 14, 15]. Furthermore, numerosity is an innate perception, very much like taste, sight, touch, smell and sound, although it is conceptually a higher order cognitive construct than the common five senses. In the absence of learning newborn human infants can respond to abstract numerical quantities across different modalities and formats [16, 17, 18]; even chicks can discriminate quantities in visual stimuli without training [19, 20].

Indeed, the tantalizing question whether neutral networks can acquire the number sense, a benchmark mental construct in neuroscience and cognitive psychology, has attracted attentions of a number of researchers. Stoianov and Zorzi were the first to study the neural network as a possible model for

---

∗Corresponding author

numerosity. They proposed a feed-forward neural network of two hidden layers to learn numerosity and tried to interpret the neurons' responses in the resulting network [21]. They reported that their neural network model exhibited a number sensing behavior. In their experiments the training samples were simple binary images depicting numbers. But the test images were drawn from the same distribution as the training images; as such, the reported results did not establish that the numerosity behaviour could generalize beyond the i.i.d. condition.

Very recently, Nasr *et al.* published another study on the numerosity capability of deep convolutional neural networks [22]. They examined a DCNN that was trained for the task of visual object recognition not explicitly for numerosity as in [21]. The training images are of natural color type, whereas the test images are binary visual representations of numbers. Somewhat surprisingly, the authors claimed that number-responsive neurons (called numerosity-selective network units) emerge spontaneously in the said DCNN for object recognition. Specifically, 9.6% of neurons of the recognition network were found to be selective to the number of objects in binary test images. Furthermore, the responses of these numerosity-selective neurons exhibited a meaningful numerosity pattern that highly resembles to those of biological neurons in monkey prefrontal cortex [23].

In the first part of this paper, we prove that the conclusion of [22] is incorrect and maintain that the pure data-driven, black box DCNN methodology cannot learn the abstract notion of numbers. Our main critique of [22] is that the number of test images used to identify number-aware neurons is two orders of magnitude smaller than the number of neurons in the network. We show that the number of so-called "numerosity-selective units" found by the ANOVA method in [22] decreases drastically as the number of test images increases. Another piece of evidence for the invalidity of the authors' claim is that, if tested with a small enough number of images, the numerosity-selective neurons can be found even in a randomly initialized network that is not trained for any task at all.

In the second part of this paper, we launch an inquiry into whether the number sense will emerge in a DCCN that is purposefully trained for cognising numbers as an abstract concept, i.e., acquiring a generalization capability to infer the number of objects in an image even if the shape, size and density of the objects differ from those of training images. To answer the question we analyze how the neurons respond to numbers presented to them in image form, if the DCNN is trained and tested for numerosity. Our analyses shed some light on the capability and limitations of DCCNs in solving cognitive problems of the simplest type, and also on the responses of the DCCN neurons in contrast to those of monkeys' neurons published by Nieder *et al.* [23].

Our findings in the second part of the paper are mixed. DCCNs can acquire a number sense through supervised learning. But this is achieved under the i.i.d. condition, i.e., only if the test and training images are drawn from the same probability distribution. The trained numerosity DCCN cannot generalize to cases where the objects in test images have minor variations from the training images; for instance, object size falls out of the range of the training images. This work exposes an uncomforting contrast between how well DCNNs can do in visual recognition tasks, some of which are quite challenging even for humans (e.g., judging if two face images are of the same person [24]), and how handicapped they become when embarking on cognitive tasks as simple and basic as learning the concept of numbers.

But on the other hand, not all results are pessimistic. At the end of the paper, we present evidences suggesting that DCNNs are more robust to distribution shift for small numbers (1 to 4) than for large numbers.

## 2   Review of Nasr *et al.*'s Work

In order to establish their hypothesis on the numerosity property of DCNNs, Nasr *et al.* [22] trained a deep convolutional neural network to classify objects in natural images using the ILSVRC2012 ImageNet dataset [25]. Their network consists of eight convolutional layers, five max-pooling layers and one fully-connected layer. The network architecture design was inspired by the discovery of simple and complex cells in early visual cortex [26, 27].

Given the trained object recognition network, the authors investigated whether varying the number of objects in a scene elicits different activations in the network units. To this end, they created a test set containing images of dot patterns depicting numbers ranging from 1 to 30 ($n = 1, 2, 4, 6, ...30$), following the experiments designed for monkeys [23] (Fig. 1 shows examples of the test images,

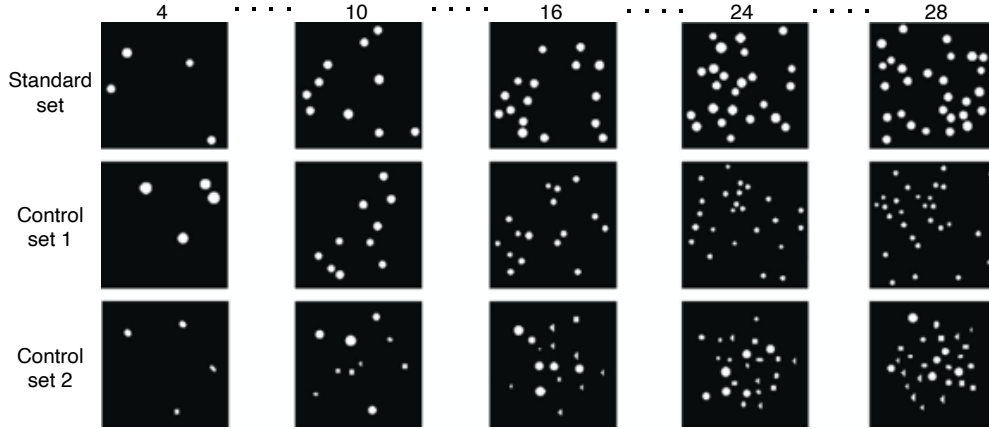

Figure 1: Sample images in the three stimulus sets. Standard set contains images displaying objects as circular dots of random size and spacing. Control set 1 contains images displaying dots of equal total area and the same dot density across numerosities. Control set 2 contains images displaying objects of different geometric shapes with an equal area of the convex hull.

called dot displays). To control for the effects that the visual appearance of the dot displays might have on unit activations, three stimulus image sets were designed. The images in the first stimulus set (standard set) display objects as circular dots of random size and spacing. The images in the second stimulus set (control set 1) display dots of equal total area and the same dot density across numerosities. The images in the third stimulus set (control set 2) display objects of different geometric shapes with an equal area of the convex hull covering all objects across numerosities. Seven images are generated for each combination of the numerosity and the stimulus set, meaning that a total of $7 \times 3 \times 16 = 336$ images are used to evaluate the responses of the network units. The sample sizes of all images and images of the same numerosity are adjusted according to the requirements of the electrophysiological monkey experiments [23].

The generated 336 test images are fed to the network and the responses of the final convolutional layer are recorded. Next, the authors perform a two-way analysis of variance (ANOVA) to detect numerosity-selective network units. Numerosity-selective units are defined to be those that generate significantly different responses across numerosities but with an invariant response across stimulus sets. Specifically, a network unit is considered to be number-selective if it exhibits a significant effect for numerosity ($p < 0.01$) but no significant effect for the stimulus set nor interaction between the two factors ($p > 0.01$).

Of the 37632 network units in the final convolutional layer, 3601 (9.6%) are found to be numerosity-selective. The responses of these numerosity-selective units exhibit a clear tuning pattern that is virtually identical to those of real neurons from monkey prefrontal cortex.

Each network unit responds maximally to a number, called its preferred numerosity (PN), and progressively decreases its response as the input number deviates from the PN. The distribution of PNs covers the entire number range (1 to 30), with more network units preferring smaller numbers, similar to the distribution observed in real neurons.

## 3   Critiques of General DCNN Numerosity

This section presents our arguments to refute the claim of [22] that training a DCNN for object recognition endows the network units with a spontaneous number sense. First, we demonstrate that the authors' conclusion is misled by an insufficient amount of test data when verifying their hypothesis. Then, we strengthen our case by showing that the same flawed data analysis could lead to an absurd conclusion: the numerosity-selective neurons exist even in a randomly initialized network that is not trained for any task at all, if this hypothesis is tested with a small enough number of numerosity images.

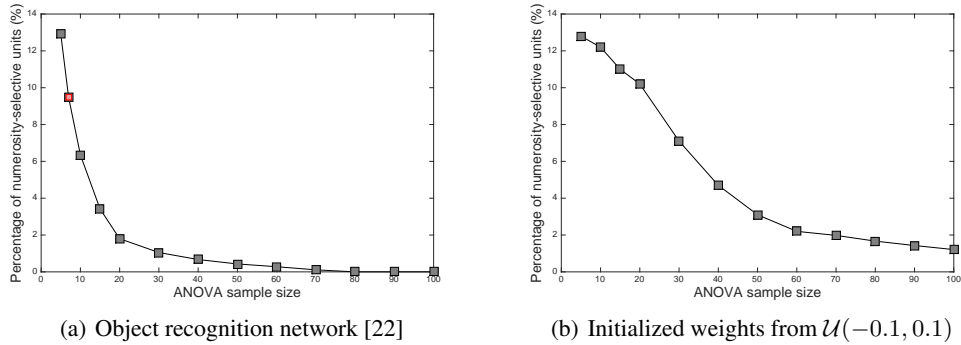

(a) Object recognition network [22]    (b) Initialized weights from $\mathcal{U}(-0.1, 0.1)$

Figure 2: The percentage of numerosity-selective neurons found by ANOVA. (a) Network trained for visual object recognition. Red point denotes the result reported in [22]; (b) Untrained network with initialized weights from a uniform distribution $\mathcal{U}(-0.1, 0.1)$.

## 3.1 Effect of expanding test set

In [22], the authors generate only seven test images for each combination of the numerosity and the stimulus set and feed them to the network, when detecting numerosity-selective network units by the two-way ANOVA method. In other words, the neuron responses were statistically analyzed with only $7 \times 3 \times 16 = 336$ images. In sharp contrast, the tested network has 37632 neurons in the final convolutional layer, three orders of magnitude greater. Given such a large gap between the sample size and the model size, their observation that 9.6% of the neurons are numerosity-selective is likely to be the outcome of small sample size instead of the manifestation of the hypothesized numerosity property.

In order to verify our suspicion, we redo the same hypothesis test on the DCNN numerosity property using the two-way ANOVA tool; only this time we gradually increase the sample size from 5 to 100 for each combination of the numerosity and the stimulus set, and examine how the percentage of neurons being numerosity-selective varies with respect to the sample size. As we suspected, the number of numerosity-selective neurons decreases drastically in the sample size of ANOVA; this trend is plotted in Fig. 2(a). In the figure the red point denotes the result (9.6%) reported in [22]. As the number of test images for each combination case increases to 20, the percentage of numerosity-selective neurons decreases to about 2%. This percentage drops to zero when the number of test images reaches 80 per combination case. Eventually no numerosity-selective neurons exist, nullifying the original hypothesis that the object recognition DCNN has the number sense.

## 3.2 Case of untrained DCNN

Now we demonstrate that the flawed data analysis of [22] can even erroneously establish the numerosity property of an arbitrary DCNN untrained for any task. We randomly initialize the same DCNN in the previous discussions with weights from a uniform distribution $\mathcal{U}(-0.1, 0.1)$, then feed the same test images as in [22] into the untrained neural network and record the neuron responses of the final convolutional layer. Somewhat surprisingly, by performing ANOVA on the neuron responses, we find a significant portion of neurons to be numerosity-selective despite the fact that the network is never trained. Specifically, 12.8% of the neurons in the untrained network are numerosity-selective, which is higher than 9.6%, the percentage in the network trained for object recognition.

Fig. 2(b) plots the effects of the sample size on the percentage of numerosity-selective neurons in the untrained network. As shown in the figure, the number of so-called numerosity-selective neurons rapidly decreases in the sample size of ANOVA, and eventually becomes too small to validate the numerosity hypothesis.

The paper [22] presented another evidence to support the hypothesized number awareness of the recognition DCNN. The authors observed that the responses of numerosity-selective DCNN neurons exhibit tuning patterns that closely resemble to those of real neurons in monkey prefrontal cortex, as by comparing Fig. 3(a) and Fig. 3(b). These number response curves are generated by pooling and

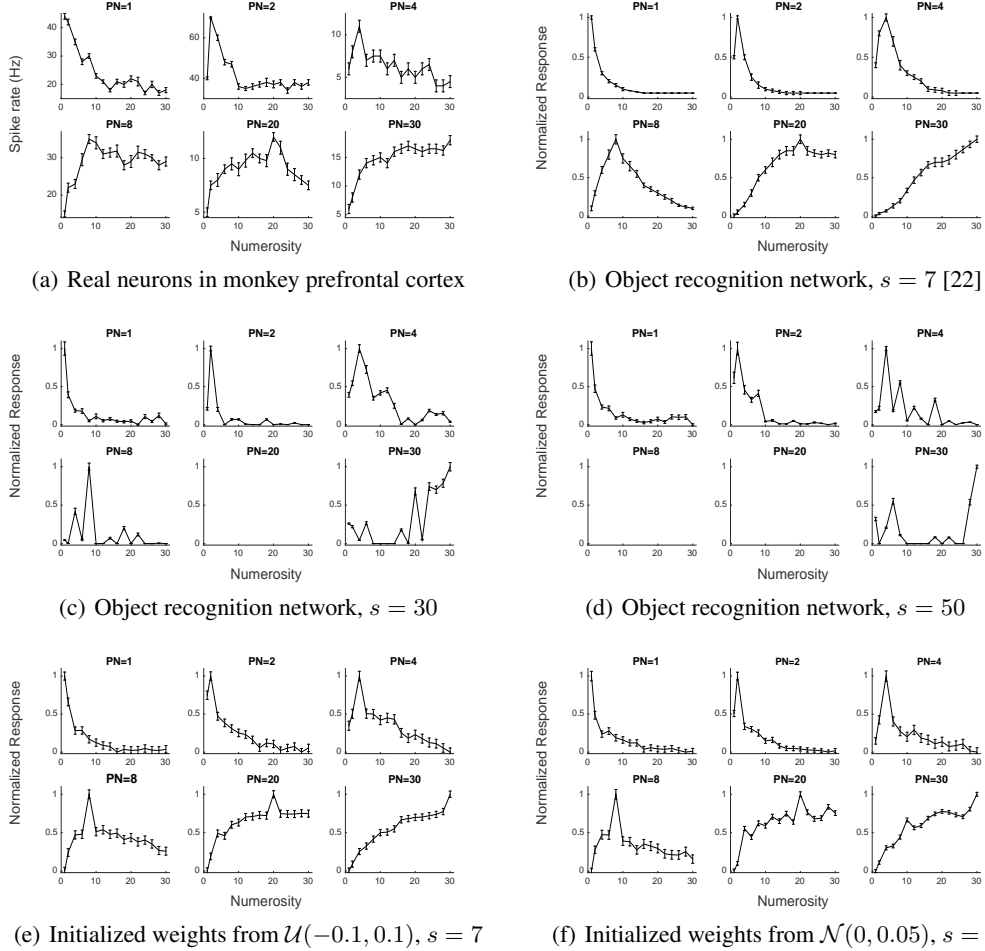

(a) Real neurons in monkey prefrontal cortex

(b) Object recognition network, $s = 7$ [22]

(c) Object recognition network, $s = 30$

(d) Object recognition network, $s = 50$

(e) Initialized weights from $\mathcal{U}(-0.1, 0.1)$, $s = 7$

(f) Initialized weights from $\mathcal{N}(0, 0.05)$, $s = 7$

Figure 3: The number response curves of numerosity-selective neurons found by ANOVA. $s$ denotes the per-case ANOVA sample size. (a) Real neurons in monkey prefrontal cortex; (b)(c)(d) Network trained for visual object recognition; (e) Untrained network with initialized weights from a uniform distribution $\mathcal{U}(-0.1, 0.1)$; (f) Untrained network with initialized weights from a normal distribution $\mathcal{N}(0, 0.05)$. Error bars indicate SE measure. PN, preferred number.

averaging the number response curves of the neurons that share the same preferred numerosity, and normalized to [0,1]. These curves are unimodal and peak at the preferred number.

Needless to say, the observed behaviour similarity between monkey neurons and DCNN neurons is also a result misled by small sample size of ANOVA. When the sample size $s$ increases, the percentage of so-called numerosity-selective neurons becomes so small that the number response curves cannot even be formed, see Fig. 3(c) and Fig. 3(d). But interestingly, the monkey-like number response patterns of "numerosity-selective" neurons also present themselves for untrained randomly initialized DCNNs if the ANOVA sample size in the hypothesis test is not sufficiently large. Figs. 3(e) and 3(f) are the number response curves of "numerosity-selective" neurons found by ANOVA using the same set of test images as in [22] but applied to the untrained DCNN of different random initializations.

## 4   Neural Network Trained for Numerosity

In the previous section, by increasing the ANOVA sample size when testing the DCNN numerosity hypothesis we reject the existence of so-called numerosity-selective neurons in the object recognition network as claimed in [22], or in untrained randomly initialized networks. The next natural question to ask is what if a DCNN is purposefully trained for the cognitive task of numerosity? This problem

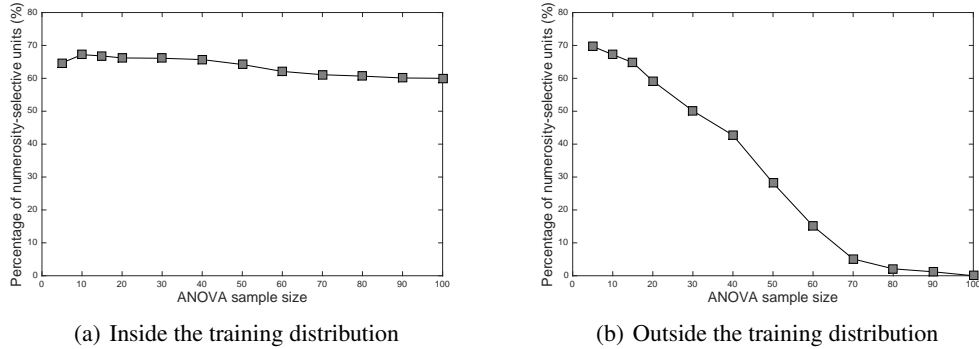

(a) Inside the training distribution        (b) Outside the training distribution

Figure 4: The percentage of numerosity-selective neurons found by ANOVA in the Nu-Net. (a) Test images are drawn from the same distribution of the training images; (b) Test images are the same as the training images but have 50% greater variations in object size.

was studied by Wu *et al.* [28]. They trained DCNNs to learn subitizing, a basic and important type of numerosity. Subitizing refers to the innate ability of humans and some animals to glance at a small set of (no more than five) objects and know how many items are in the set. The findings of [28] contradict the key claim of [22] and expose the inability of data-driven black box DCNNs to learn the abstract concept of numbers. The authors showed that DCNNs could not be trained, even with strong supervision, to perform subitizing with human-like generalization power.

The investigative approach of [28] is largely functional, making conclusions based on a family of cognitive psychology experiments. In contrast, we are interested in analyzing how the neurons respond to numbers presented to them in image form, if the DCNN is trained for numerosity. What follows are our efforts to understand the capability and limitations of the DCNN trained for numerosity and to interpret the inner works of the neurons.

In cognitive science, numerosity is a raw perception rather than resulting from arithmetic, accordingly we model it as a problem of classification rather than regression. We train a numerosity DCNN, denoted by Nu-Net, by teaching it to perform the 16-label classification task. The 16 output labels correspond to natural numbers 1, 2, 4, 6, 8, 10, 12, 14, 16, 18, 20, 22, 24, 26, 28, and 30. We let the Nu-Net have the same network architecture as in [22] for this architecture is biologically inspired [27] and for fair comparison later. The number-depicting images (see Fig.1) for training the Nu-Net are the same as in [22], which are also used in the numerosity study on monkeys [23]. Specifically, two hundred images are generated for each combination of the numerosity and the stimulus set, meaning that a total of $200 \times 3 \times 16 = 9600$ binary images depicting numbers are used to train the Nu-Net. We also generate a test image set containing one hundred images for each combination of the numerosity and the stimulus set, in the same way as the training image set. Thus, the images in the training set and test set are from the same distribution.

Given the past successes of DCNNs in solving visual classification problems, it is not surprising that the Nu-Net infers the number of objects almost perfectly (95.6% accuracy) on test images that are drawn from the same distribution of the training images. In order to interpret the high accuracy we record the neuron responses of the final convolutional layer when the Nu-Net reads in the number-depicting test images. By performing the two-way ANOVA on the neuron responses, we find that 68.4% of the neurons are numerosity-selective. That is, these neurons' responses are significantly different across numerosities but with an invariant response across stimulus sets, i.e., independent of object shape, size and density. Unlike in the case of object recognition network, here the percentage of numerosity-selective neurons hardly changes in the per-case ANOVA sample size $s$. Even when $s = 100$, there are still about 60% of neurons that are numerosity-selective (see Fig. 4(a)). In addition, the number response curves of these neurons closely resemble those of real neurons in monkey prefrontal cortex, as shown in Fig. 8(a).

The above results are not surprising, because the Nu-Net is trained specifically for numerosity and the test images are drawn from the same distribution of the training images. Neural networks are expected to succeed in making statistical inferences under the i.i.d. condition. But does this mean

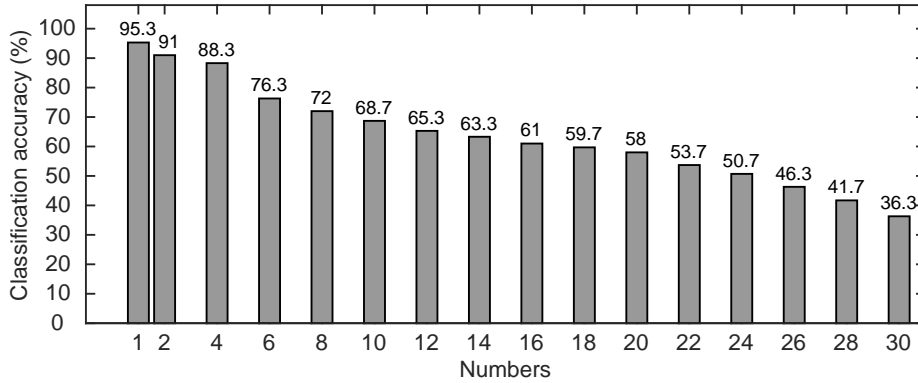

Figure 5: Classification accuracy of the Nu-Net for each number on the test images that have 50% greater variations in object size than the training images.

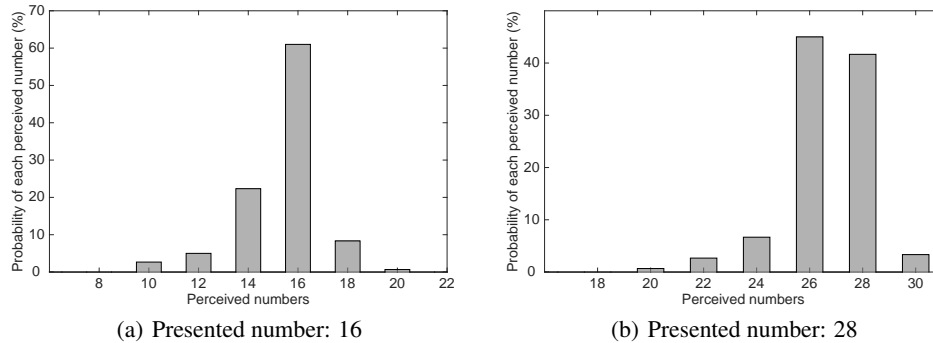

(a) Presented number: 16

(b) Presented number: 28

Figure 6: Two typical distributions of the perceived numbers of the Nu-Net.

that the Nu-Net has learnt the abstract notion of numbers like preschoolers do, i.e., can its accuracy in numerosity be maintained across objects of shape, size and density unseen in the training images? To answer the question, we have extensively tested the Nu-Net using number-depicting binary images of objects of greater variations in size, shape and density, and found the performance of the network on numerosity tasks deteriorates significantly.

Now, as a generalization study, we discuss our experimental results on test images that are the same as the training images but have 50% greater variations in object size. This modestly increased variability in object size decreases the average classification accuracy of the Nu-Net from 95.6% to 64.2%. Fig. 5 plots the classification accuracy of the Nu-Net for each input number. This accuracy is high up to 4, then it drops as the number of objects in test images increases. As the number of objects in test images increases the uncertainty in the perceived number by the Nu-Net increases. The distributions of the perceived numbers of the Nu-Net with input numbers being 16 and 28 are shown in Fig. 6(a) and Fig. 6(b). This unimodal distribution is typical for other input numbers.

To better characterize the accuracy of the Nu-Net in the generalization test, we compute the 85% estimation interval length for each input number. Given the presented number $x$, the 85% estimation interval length is defined to be the minimal $\delta + \epsilon$ such that $p(x - \delta, x + \epsilon) > 85\%$, where $p$ is the probability distribution of the perceived numbers. The 85% estimation interval length for each input number is shown in Fig. 7. It can be seen that, for number 1,2 and 4, the 85% estimation interval length is 1, meaning that the Nu-Net performs very well on small numbers, a property called subitizing to be discussed shortly. Starting from 6, the 85% estimation interval length increases, indicating an increasing level of uncertainty in number perception for large populations very much like humans.

In order to explain the performance deterioration caused by immaterial variations in object size in input images, we examine if and how such changes adversely affect the percentage of numerosity-

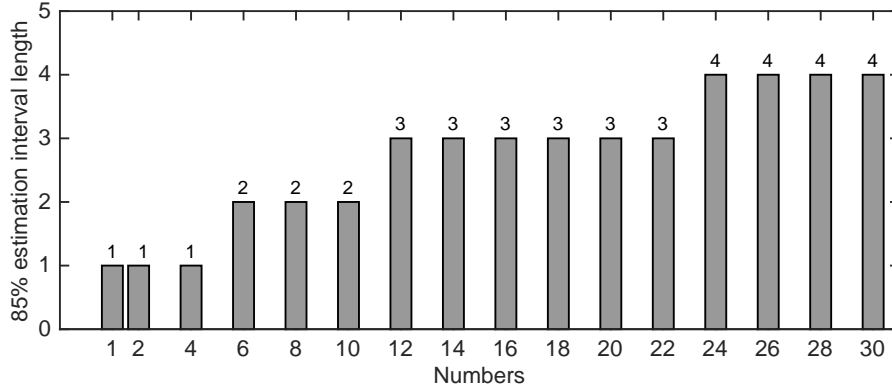

Figure 7: 85% estimation interval length for each number.

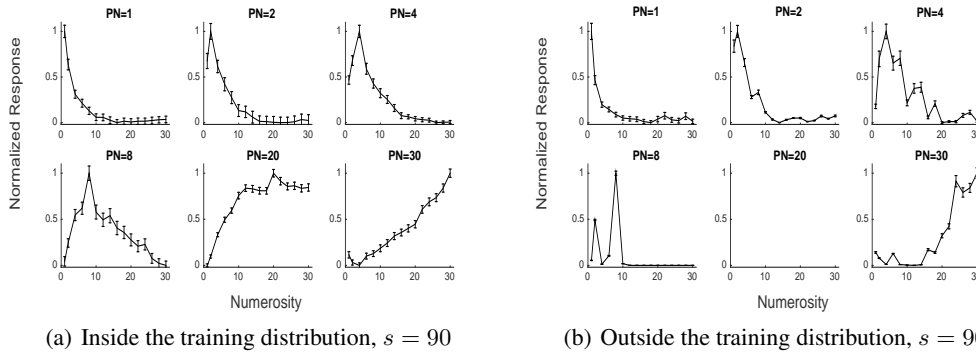

(a) Inside the training distribution, $s = 90$        (b) Outside the training distribution, $s = 90$

Figure 8: The number response curves of numerosity-selective neurons of the Nu-Net. $s$ denotes the per-case ANOVA sample size. Error bars indicate SE measure. PN, preferred number.

selective neurons and their number specificity. We test the numerosity hypothesis on the Nu-Net by analyzing the neuron responses to the superficially varied input images with the two-way ANOVA tool. As the per-case sample size $s$ of ANOVA increases from 5 to 100, the percentage of numerosity-selective neurons decreases steadily, as shown in Fig. 4(b). This percentage drops to zero when the ANOVA sample size reaches 100 per combination case. When the Nu-Net reads in test images outside the distribution of training images, not only the number of numerosity-selective neurons drops drastically, their numerosity response patterns also disappear, as shown in Fig. 8(b).

Finally, we would like to conclude the paper with a more positive note. As shown in Fig. 5, the Nu-Net has very high accuracy in numerosity tests when being presented with small numbers 1, 2 and 4 encoded in image signals outside the training distribution, despite its deteriorating performances for larger numbers in the same generalization tests. This limited success in generalized inference can be explained by examining the response curves of the neurons that are selective to small numbers 1, 2 and 4 (see Fig. 8(b)). These curves are generated by analyzing neurons' responses using two-way ANOVA when the Nu-Net reads in images of object size outside of the range of the training set, and they resemble those discovered in the numerosity experiments with monkeys [23]. In other words, supervised learning make these DCNN neurons sensitive and discriminative to small numbers, producing a dominant peak response at the preferred number. The above observations suggest that the Nu-Net has better generalization ability for small numbers (1 to 4) than for large numbers.

Perhaps more intriguingly, even for the object recognition network that is not trained for numerosity, the neurons selective to 1, 2 and 4 found by two-way ANOVA also have discriminative response curves similar to those for the Nu-Net (compare Fig. 3(d) and Fig. 8(b)). Further research on this DCNN behaviour seemingly related to numerosity is warranted.

## Broader Impact

This research contributes to the knowledge on the strengths and limitations of deep learning; particularly so for cognitive computing, considering that numerosity, together with language, is a hallmark of human intelligence.

## Acknowledgment

The authors are grateful for the fruitful discussion with Xiao Shu and Fangzhou Luo. This research was partly supported by Natural Sciences and Engineering Research Council of Canada (NSERC) and SJTU Overseas Study Grant.

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
