[Supplementary Material]

# Supplementary Material

## On Numerosity of Deep Neural Networks

## 1    Generalization study on object density

In the main paper, due to the page limit, we only discussed the beyond i.i.d. generalization capability of the Nu-Net when object size is outside of the distribution of the training set. Here we add the generalization results of the Nu-Net when the object density is outside of the distribution of the training set. By comparing the following results with their counterparts in the main paper, one can find them to be very similar; namely, the generalization capability of the Nu-Net is about the same on object size and on object density.

Specifically, we run the Nu-Net on test images that are the same as the training images but have 50% greater variations in object density. As in the case of object size generalization, this modestly increased variability in object density decreases the average classification accuracy of the Nu-Net from 95.6% to 55.8%. Fig. 1 plots the classification accuracy of the Nu-Net for each input number. This accuracy is high up to 4, then it drops as the number of objects in test images increases. The uncertainty in the perceived number by the Nu-Net increases in the number of objects in the test image. The distributions of the perceived numbers of the Nu-Net with input numbers being 16 and 28 are shown in Fig. 2(a) and Fig. 2(b), respectively.

Figure 1: Classification accuracy of the Nu-Net for each number on the test images that have 50% greater variations in object density than the training images.

(a) Presented number: 16

(b) Presented number: 28

Figure 2: Two typical distributions of the perceived numbers of the Nu-Net.

The 85% estimation interval length for each input number is shown in Fig. 3. It can be seen that, for numbers 1,2 and 4, the 85% estimation interval length is 1, meaning that the Nu-Net performs very well on small numbers, i.e., on the task of subitizing. Starting from 6, the 85% estimation interval length increases, indicating an increasing level of uncertainty in number perception for large populations very much like humans.

Figure 3: 85% estimation interval length for each number.

As shown in Fig. 4(b), when the Nu-Net reads in images of object density outside of the range of the training set, the number response curves of numerosity-selective neurons of the Nu-Net are still selective to small numbers 1, 2 and 4, just like those discovered in the generalization study on object size. This observation supports the conclusion made in our paper that the Nu-Net passed the generalization tests on subitizing, namely, judging the number of objects in an image with confidence and robustness, as long as that number is 4 or smaller.

(a) Generalization on object size, $s = 90$

(b) Generalization on object density, $s = 90$

Figure 4: The number response curves of numerosity-selective neurons of the Nu-Net. $s$ denotes the per-case ANOVA sample size. Error bars indicate SE measure. PN, preferred number.

## 2 Inference on real-world images

The trained Nu-Net can also be used to count objects in real-world images, with a simple pre-processing step to convert real-world color images to binary images. Fig. 5 shows some examples of the real-world images and the corresponding binary images after pre-processing. The Nu-Net infers the number of objects almost perfectly on this type of real-world images.

Figure 5: Left: real-world images. Right: binary images after pre-processing.