[Reviews · NeurIPS 2020]

Review 1

Summary and Contributions: The authors debunk a finding in the literature which reported that a CNN trained for object recognition contains neurons which are sensitive to numerosity (e.g. the feature of having seven objects in the image.) They next train a network specifically to learn numerosity, but find that it fails to generalize out-of-distribution, except for numbers 1, 2, and 4.

Strengths: The claims of the paper are solidly established with appropriate statistical methodology. The idea of using a randomly initialized network as a control is clever and, in conjunction with other analyses, leads to a convincing case against the claims in the previous literature.

Weaknesses: The paper is primarily descriptive and does not attempt to say too much about the theoretical aspects. It is slightly dissatisfying that only 16 number classes were used (but perhaps this was to imitate previous work); in particular, that there is no class for the number 3. The application to broader datasets is questionable. Like previous works, the number detecting task is rather artificial. Although successful application to real-world datasets is shown in the supplement, it relies on a relatively severe form of pre-processing.

Correctness: The methodology is correct. The authors appropriately use tests that differentiate between in-sample generalization, out-of-sample generalization and out-of-distribution generalization.

Clarity: The paper is clearly written, and its significance in the larger context of machine learning and neuroscience is also clearly conveyed.

Relation to Prior Work: Yes.

Reproducibility: Yes

Additional Feedback: Update after rebuttals: It is reassuring that the results still hold for number classes 1-30, but I wish the authors were more specific about the obtained results in the rebuttal. The fact that the experiments were targeted at showing an upper bound on performance (using unrealistic and favorable stimuli) further reaffirms the comments by other reviewers that the positive claims in the paper should be downstated. In light of the opinions of the other reviewers who are more knowledgable about this area than I am, I am lowering my score from 8 to 7 and also lowering my confidence from 4 to 3. Original: ---------- It would make the work more impactful if Nu-Net could be trained on a more realistic dataset.


Review 2

Summary and Contributions: Edit after author rebuttal and reviewer discussion: I've read the author's rebuttal in which they have addressed several of my concerns. I maintain that the contribution is important and that the paper should be published. However, I also think that reviewer 4 has a good point about not overselling the positive results. I encourage the authors to temper their discussion of their evidence for subtilizing to not over-generalize or exaggerate the results. _______________________________________________________________________ This submission responds directly to previously published work (Nasr et al 2019) claiming the emergence of number sense, the ability to perceive numerosity, in neural networks trained only to recognize visual objects. The authors demonstrate that the results in Nasr et al are an artefact of the sample size and appear even in random, untrained networks. These results do not persist when using larger sample sizes. The authors then go on to train neural networks explicitly for numerosity. In general, they found that networks are only able to generalize to images within the distribution they were trained on and do not learn an abstract concept of numbers that generalizes to new data distributions. However, they do provide some evidence for the ability to subitize (the ability to glance at a small set of objects and know how many are present) by demonstrating out-of-distribution generalization for very small numbers (1–4).

Strengths: * The Nasr et al paper to which this submission responds has been published in a high impact journal and has already been cited several times. As such, it seems imperative to publish this work demonstrating the methodological flaws of that work which completely reverse the results. I am personally not surprised that the neural networks tested do not learn the abstract concept of number, but it is none the less important to have an empirical paper showing this to be so to counter the notion that such an abstract concept could emerge in a network trained only to recognize objects. The result will certainly be relevant to the NeurIPS community, especially those working at the intersection of deep learning and cognitive science. * I was also struck by the result that even random untrained networks show tuning curve-like number selectivity. It makes me wonder how confident we are that the tuning curves we measure from biological neurons are relevant to numerosity. This result will likely be of interests to neuroscientists as well. * Analyzing individual unit response properties allows for the comparison to responses recorded from individual biological neurons

Weaknesses: * Both this paper and the Nasr et al paper use number detectors (number selective units) as an indicator of number sense. However, the presence of number selective units is not a necessary condition for number sense. There are potentially other distributed coding schemes (other than tuning curves) that could be employed. It seems like the question that you really want to ask is whether the representation in the last convolutional layer is capable of distinguishing images of varying numerosity. In which case, why not just train a linear probe? Number sense is a cognitive ability, not a property of individual neurons. We don't really care what proportion of units are number selective as long as the network is able to perceive numerosity (which might not require very many units). A larger proportion of number selective units doesn't necessarily imply a better number sense. As such, I question the reliance on the analysis of individual units and would rather see population decoding results. * The motivation for analyzing only the last convolutional layer is not clear. Why would numerosity not appear in earlier layers? * The motivation for using classification rather than regression when training explicitly for numerosity is not well justified. The justification, "numerosity is a raw perception rather than resulting from arithmetic", is not clear. Humans clearly perceive numbers on a scale not as unrelated categories. That the subjective experience of numerosity does not involve arithmetic does not constrain the neural mechanisms that could underly that perception. * No effect sizes are reported for number selectivity. Since you did ANOVAs there should be an eta squared for the main effect of numerosity. How number selective are these units?

Correctness: * No correction for multiple comparisons was reported, despite the large number of hypothesis tests performed. Given the alpha of 0.01, 1% of the identified numerosity selective units are likely false positives. This does not affect the conclusions of the paper but would be good to mention.

Clarity: * There are several sentences that are slightly unclear and the paper needs to be proof read for typos (I’ve identified several specific small edits below). However, in general, the motivation, goals, methods and results of the paper are clear. * Figure 3 and Figure 8: Both figures have the same problems. The labels and titles are much too small to read. I don’t see why you need to plot every line in its own plot. Each group of six plots could be combined into a single plot with six lines of different colors and a legend to indicate the PN associated with each color. Instead of the error bars, you could draw a partially transparent band around each line which would help to declutter the multiline plot. Minor issues: * line 39: are you missing the “feed” in “feed forward neural network”? * line 62: do you mean “inquiry” instead of “enquire”? * line 70: not clear what you mean by “Our findings in the second part of the paper are fixed”. what do you mean “fixed”? * line 125-132 (first paragraph of 3.1): * there are a few issues in this paragraph. You write “as many as” and “as few as” which suggests a variable number when I think you mean to state an exact number. If so, just remove “as many as “ and “as few as”. * Missing comma after “In sharp contrast" * The first sentence doesn’t read well, consider rewording * line 176: missing the word “at” in “to glance (at) a small set" * line 182: replace “experiments of cognitive psychology type” with “cognitive psychology experiments”. There are a number of phrases like this “noun of noun type” which seem a bit odd. * line 223: missing comma after “increases" * line 236-237: extra “the” and extra “on”. i.e. “affect the percentage of () numerosity selective neurons and () their number specificity" * line 243: “also disappear" should go at the end, i.e. “their numerosity response patterns also disappear.

Relation to Prior Work: It is clear how this work builds on the closely related work of Nasr et al [22] and Wu et al [28]. I am not an expert in numerosity so there could be other relevant literature that I am unaware of.

Reproducibility: Yes

Additional Feedback:


Review 3

Summary and Contributions: This paper analyzes CNNs to determine whether internal neurons encode abstract numerosity. The work examines previous research claiming that a CNN trained on natural images represents numerosity. The present work exposes methodological flaws in this earlier research by varying the ANOVA sample size and showing that an untrained network behaved much the same as a trained network. The paper goes on to analyze a CNN trained on a numerosity task. It finds that while the network does not generalize well to OOD examples, it does seem to show the ability to subitize (apprehend numerosity of 1-4 elements). Authors find that a network trained to perform numerosity judgment has neurons in the last convolutional layer that encode numerosity. Not a terribly surprising result (as the authors themselves state), except insofar as numerosity is encoded in single neurons versus activity vectors.

Strengths: The paper is very well written and the experiments are straightforward. I am convinced by the paper's take-down of the earlier flawed work [ref 22]. The indicators of robust subitizing is intriguing.

Weaknesses: This work is mostly an analysis of previous studies of numerosity, with a few small extensions. The previous work showed via behavioral measures that networks trained to subitize did not exhibit human-like generalization power; this work makes a similar argument by examining internal neuron responses. Although the work is a solid experimental study, the results are mostly negative and the lack of algorithmic novelty will limit its appeal to the NeurIPS audience. My biggest disappointment with the empirical work is that the OOD numerosity tests for Nu-Net seemed to be using a distribution not terribly distinct from the training distribution. I would have liked to have seen results from radically different numerosity images. The human subitizing literature indicates robustness to the elements being enumerated (e.g., natural images with a varying number of tokens). Before claiming that the net can subitize, robustness should be tested.

Correctness: The work appears to be carefully conducted and analyzed.

Clarity: The paper is very well written and clear.

Relation to Prior Work: The connections to the literature are well stated.

Reproducibility: Yes

Additional Feedback: 47-48: unclear what you mean by "binary visual representations of numbers" CNNs have strong inductive bias, as evidenced by Zhang (https://arxiv.org/abs/1902.04698) and Ulyanov (https://arxiv.org/abs/1711.10925). Is it completely unreasonable to suppose that the architecture of CNNs could, in and of itself, support some sort of numerosity estimation? The authors claim that their Nu-Net performs subitizing, yet the numbers in question are small (1, 2, and 4) the 85% estimation interval is 1. Does this mean that there's uncertainty as to whether the actual number was 1 more or less? I believe the authors are saying that \delta+\epsilon=1 but that leaves ambiguity whether off by 1 errors can occur, which is not what happens with subitizing. I would recommend that the authors sell their work as a negative result, namely that (1) the original work of [22] did not find evidence for subitizing or numerosity estimation in a CNN, and (2) even when the net is trained to report numerosity, it does not learn an abstract concept of number because it does not show transfer to different types of objects.


Review 4

Summary and Contributions: Update after author response: I would like to thank the authors for the detailed response. I am changing my score from 4 to 5 based on the reviewer discussion and the author response. Some more detailed thoughts follow. While I agree that the negative results here are quite interesting and a step towards slowing the trickle of overreaching conclusions, I am still unconvinced about the positive results. Essentially, a network trained for a simple task does well on the task, and fails to generalize in a robust manner. This is a very expected result in my opinion. The out of distribution samples only have "50% greater variations in object size" and yet the performance deterioration is substantial (from 95.6% to 64.2%). The other control stimuli (Fig. 1) have been completely ignored in this analysis without any reason or discussion. The conclusion made -- emergence of subitizing behavior, is overreaching in a similar fashion to the work the authors set out to debunk. How can we have any confidence that just changing the shape of the stimulus from dots to triangles won't change this conclusion? What about changing density? Size? Noise levels? Inverting the color-map? And so on. The rebuttal continues to argue the significance of the positive findings ("We did show CNNs can learn subitizing with good accuracy and robustness"), but I don't agree that the results are robust if the only manipulation (change in size) already lead to significant performance degradation. Similarly, the claims on the parallels with primate cognition appear unsubstantiated ("Our partially positive finding points to an intriguing computational parallel to the innate capability of subitizing of humans and primates."). What are these parallels? That brains implement some form of CNN? Same behavioral observations can emerge from very many different hardware implementations, and doesn't form grounds for these claims. If the paper largely sticks with the singular theme of disproving the results of a previously published paper in a reputable venue, it would make more sense. As such, the message is mixed and diluted with the paper falling into the same trap it sets out to warn the scientific community about -- making overly general claims from a small set of experiments. I would really like to see a follow-up paper with more extensive experiments that ground these claims, making them indisputable. ------- In this paper, the authors disprove a claim made by Nasr et al. that numerosity judgement can spontaneously emerge in neural networks (NN) trained for an object recognition task. They show the observation to be an artifact of the small number of samples used to substantiate the claim, and furthermore that a similar behavior can potentially emerge in NN with random weights. The authors then train a new NN on the specific task of numerosity judgement showing that while it performs well for samples coming from the same distribution as training data, its performance deteriorates when a different data distribution is used. The one silver-lining being the preservation of desired tuning behavior for the neurons responsive to smaller preferred numbers.

Strengths: The paper is well-organized and easy to read, and highlights the importance of caution and deeper scrutiny with ambitious claims.

Weaknesses: I found most of the results of the paper to be quite unsurprising. Unfortunately in this field, there are many papers that cannot be reproduced or make claims that stretch beyond what’s scientifically warranted. A follow-up paper showing the claims do not hold is only of limited interest, even when accompanied with good analysis. Albeit somewhat reductionist, the contribution of the paper centers around the well-known issue of lack of generalization across different data distribution, just elaborated for a specific case. Ironically, the claim in Broader impact is somewhat overreaching itself, “numerosity, together with language, is a hallmark of human intelligence.” Minor grammatical issues like, “launch an enquire”.

Correctness: Yes.

Clarity: Yes.

Relation to Prior Work: Yes.

Reproducibility: Yes

Additional Feedback:

[Author Response · NeurIPS 2020]

**Response to Reviewer #1**

- *"It is slightly dissatisfying that only 16 number classes were used; there is no class for the number 3."*
- We reported only 16 number classes to match the experiments of Nasr *et al.*'s work [22]. We DID the hypothesis tests for all 30 number classes (from 1 to 30) and the findings are consistent with those of 16 number classes.

- *"It would make the work more impactful if Nu-Net could be trained on a more realistic dataset."*
- Using binary abstract number-depicting images is the tradition of numerosity studies in cognitive sciences. We trained and tested with more realistic images and the accuracy is worse than abstract images. The paper focuses on the basic science problem not engineering applications. Nevertheless, if CNN fails to generalize on simplistic examples, let alone far more varied practical cases.

**Response to Reviewer #2**

- *"Number sense is a cognitive ability, not a property of individual neurons."*
- We appreciate this reviewer's argument for the possibility of a distributed coding scheme for the number sense. Numerosity cognition may well be a holistic mechanism. Like this reviewer we had the same urge to challenge the methodology and results of numerosity studies in neuroscience [21][22], but refrained from debunking directly the works published in top journals. Thanks to your insight and support, we will make this point in the final version.

- *"The motivation for analyzing only the last convolutional layer. Why would numerosity not appear in earlier layers?"*
- Empirical studies show that deeper layers in CNN encode higher level concepts than shallower layers. Front CNN layers extract low level features (e.g., corners, edges, textures, etc.) Semantics tends to emerge from deep layers. Numerosity, as an abstract cognitive concept, should be exhibited by very deep layers. This is why we and previous authors only examined the last convolutional layer. But for the sake of thoroughness we will check all layers and discuss the results in the final version.

- *"The motivation for using classification rather than regression is not not well justified."*
- We followed the well accepted belief that subitizing is a raw perception not resulting from deliberate calculation. In fact, we also tested regression formulation, the results hardly changed.

- *"No effect sizes are reported for number selectivity."*
- The average $\eta^2$ for the numerosity effects of all number-selective units decreases from 0.25 to 0.08, when the sample size increases from 5 to 100. We will add the effect sizes in the final version, as suggested.

- *"Figure 3 and Figure 8 ... labels and titles are much too small".* - Thanks, will improve as suggested.

**Response to Reviewer #3**

- *"I would have liked to have seen results from radically different numerosity images ... "*
- We did train and test on numerosity images of much greater variations, and found the inference accuracy and robustness of subitizing decrease. We didn't include these results to keep our experiments in the same setting as in the previous numerosity studies. We will add discussions on more varied sample images in the final version as suggested.

- *"unclear what you mean by 'binary visual representations of numbers' "* - Black and white images depicting numbers.

- *"completely unreasonable to suppose that the architecture of CNNs ... support some sort of numerosity estimation?"*
- We guess here you doubted if CNNs can learn subitizing beyond i.i.d. inference. Indeed, Zhang's work shows empirically that CNNs have the ability to generalize beyond the training images in the identity-mapping task, even trained on a single example. Why shoudn't it be possible for CNNs to succeed in the task of subitizing.

- *"Authors claim that Nu-Net performs subitizing, yet the small numbers (1, 2, 4) the 85% estimation interval is 1."*
- Thanks for pointing out the error. For subitizing the 85% estimation interval length is 0, NOT 1, i.e., $\delta$ and $\epsilon$ are both 0 when $x < 5$. Likewise, the height of all bars in Fig.7 should be reduced by 1. Nu-Net makes no errors in subitizing more than 85% of times; off by 1 errors can occur but with less than 15% chance. We will fix the errors and clarify.

**Response to Reviewer #4**

- *"A followup paper showing the claims do not hold is only of limited interest, even accompanied with good analysis."*
- In terms of neuroscience, our negative results are fascinating and have far reaching implications by exposing a pitfall of a standard methodology in published studies of biological neurons; that is, identify number selective neurons via ANOVA. As pointed out by reviewer 2, it is "imperative" to publish these findings, because our critique necessitates reexaminations and calls for new understandings of numerosity, which is of importance in both AI and neuroscience.

We'd like to stress that this work is more than just negating previous well-accepted results; it also offers an interesting constructive result. We did show CNNs can learn subitizing with good accuracy and robustness, although the general numerosity problem turns out much harder. Our partially positive finding points to an intriguing computational parallel to the innate capability of subitizing of humans and primates.

[Meta-Review · NeurIPS 2020]

This paper demonstrates that an analysis relied upon in a previous paper (Nasr et al., 2019) to identify number-sensitive units in a neural network trained for object recognition is flawed, and that indeed the same network with randomly initialized weights also has a large number of number sensitive units. Moreover, the number of units detected depends strongly on the sample size of the statistical test, with larger sample sizes detecting no number sensitive units. The paper additionally performs some analyses on a network trained specifically to predict number. The reviewers generally felt that the demonstration of Nasr et al.’s flawed analysis was important, with R2 arguing that the work is “imperative to publish” and R1 and R3 finding the experiments in the first part of the paper convincing. However, R1, R3, and R4 all had concerns with the second part of the paper, in which it is claimed that a network trained to classify number (Nu-Net) can learn to subitize. I feel that the results in the first part of the paper are sufficiently impactful that the paper should be accepted. Echoing R2, I believe that the results will be of interest to both computer vision researchers and neuroscientists in terms of painting a cautionary tale against relying too heavily on the specificity of individual neurons. However, I find myself also agreeing with R1, R3, and R4 that the second part of the paper feels overclaimed. I think the analyses are interesting and thought-provoking, but I’m unconvinced that they show the network has truly learned to subitize. So, while I am recommending acceptance, I would like to see some changes to the claims regarding subitizing. For example, rather than saying “we present evidence suggesting that DCNNs can learn subitizing” or “Nu-Net has passed the generalization tests on subitizing”, it would be more accurate to state that (the specific network used) is more robust to distribution shift for small numbers than for large numbers. To make a stronger claim that NNs actually learned to subitize would require extensive experiments on many architectures with many types of datasets, which has not been demonstrated in this paper.